# Shedding a Light on Acyclovir Pharmacodynamics: A Retrospective Analysis on Pharmacokinetic/Pharmacodynamic Modelling of Acyclovir for the Treatment of Varicella Zoster Virus Infection in Immunocompromised Patients: A Pilot Study

**DOI:** 10.3390/pharmaceutics14112311

**Published:** 2022-10-27

**Authors:** Geeske F. Grit, Anne-Grete Märtson, Marjolein Knoester, Marlous L. Toren-Wielema, Daan J. Touw

**Affiliations:** 1Department of Clinical Pharmacy and Pharmacology, University Medical Center Groningen, University of Groningen, 9713 GZ Groningen, The Netherlands; 2Antimicrobial Pharmacodynamics and Therapeutics, University of Liverpool, Liverpool L69 3BX, UK; 3Department of Medical Microbiology and Infection Prevention, University Medical Center Groningen, University of Groningen, 9713 GZ Groningen, The Netherlands

**Keywords:** pharmacokinetics, pharmacodynamics, acyclovir, PKPD modeling, varicella zoster virus

## Abstract

Background: Acyclovir and valacyclovir are used for the treatment and prophylaxis of infections with herpes simplex virus (HSV) and varicella zoster virus (VZV). The aim of this study is to provide insight into the pharmacodynamics (PD) of (val)acyclovir. Methods: Patients were retrospectively selected, based on therapeutic drug monitoring for acyclovir, to create a population pharmacokinetic (PK) model in Pmetrics. This PK model was used to develop a PK/PD model to study the effect of acyclovir levels on VZV viral load in plasma in immunocompromised patients. Results: Immunocompromised patients with known VZV viral loads in plasma were included for PK/PD modelling (*N* = 4, with 23 measure points); they were part of the population of 43 patients used for PK model building. The PK/PD model described the data well (r^2^ = 0.83). This is a hopeful first step in clarifying the pharmacodynamics of acyclovir; however, the data in this study are limited. Conclusions: Our preliminary PK/PD model can be used in further research to determine the effect of acyclovir levels on VZV viral load.

## 1. Introduction

Acyclovir (ACV) and valacyclovir (VCV) are antiviral agents used for treatment and prophylaxis of infections with herpes simplex virus (HSV) and varicella zoster virus (VZV) [1,2]. VCV is a prodrug of ACV and has improved oral bioavailability, compared to oral acyclovir [2]. The susceptibility to ACV is highest for herpes simplex virus type 1 (HSV-1), followed by herpes simplex virus type 2 (HSV-2) and VZV [3]. Susceptibility is defined as 50% inhibitory concentration (IC_50_) or as 50% effective concentration (EC_50_), indicating the in vitro ACV concentration, which reduces plaque numbers or viral copies by 50%, compared to control-infected cells [4,5]. The in vitro IC_50_ of ACV for HSV-1 and HSV-2 varies between 0.02–0.9 and 0.03–2.2 mg/L, respectively. The in vitro IC_50_ for VZV ranges from 0.8 to 4.0 mg/L [3].

Currently, it is unclear which pharmacodynamic (PD) target should be achieved for effective treatment with ACV or VCV. Various studies used different PD targets [6,7,8,9,10,11], e.g., ACV concentration 50% of time > IC_50,_ or minimal ACV concentration (C_min_) > IC_50_ [6,7]. The referred studies do not provide clear evidence to support these PD targets [12,13,14,15]. However, it is assumed that the efficacy of ACV is related to time-dependent killing [16].

This study aims to provide insight into the pharmacodynamics of ACV. A population (pop) PK model was created to predict ACV levels, and a PK/PD model was developed to study the effect of ACV levels on plasma VZV viral load.

## 2. Materials and Methods

### 2.1. Population Selection

Patients of the University Medical Center Groningen (UMCG), Groningen, The Netherlands, treated with oral VCV and/or intravenous (IV) ACV for an HSV or VZV infection and who had measured ACV levels between 1 January 2018 and 1 December 2021, were included in this retrospective study. Patients on dialysis and/or who had objected to the use of their medical data were excluded from analysis. The Medical Ethics Review Board of the UMCG declared that this study did not need Medical Research Involving Human Subjects Act (WMO) approval (METc2021/029).

### 2.2. ACV Levels and VZV Viral Loads Analysis

The serum concentration of ACV was measured, as described previously [17]. The analysis of the VZV viral loads in plasma is based on Hawrami et al. [18] and is included in the Appendix A. The measured VZV viral loads (cycle threshold (Ct) values) were converted to log_10_ viral copies/mL. The calibration line of the Epstein–Barr virus (EBV) was used, since no calibration line of VZV was available. EBV was chosen because this is also a human herpes virus. The viral load was considered undetectable when no signal was found or Ct value was > 39.99. By this, the lower limit of detection was 1.4 log_10_ copies/mL. 

### 2.3. PK Model

A pop PK model of (val)acyclovir was developed and fitted to our patient data (method included in the Appendix A). The models were analysed by comparing goodness-of-fit plots, r^2^ of individual and population predictions, Akaike information criterion (AIC), Bayesian information criterion (BIC) and −2 Log likelihood values. ACV levels < 0.1 mg/L and levels with unclear ACV administration times were excluded.

### 2.4. PK/PD Model

A PK/PD model was developed in Pmetrics, version 1.9.7. for RStudio (version 1.4.1106, Laboratory of Applied Pharmacokinetics and Bioinformatics, Los Angeles, CA, USA) [19]. The model code is provided in the Appendix A. Of our population, immunocompromised patients (chemotherapy, under immunosuppressants, hematologic disorders) with at least two positive VZV viral load measurements in plasma were included in the PK/PD model. The individual posterior prediction PK parameters were calculated from the final PK model and were used as covariates for creating the PK/PD model.

The base PK/PD model was as follows:(1)$$\frac{\Delta X_{1}}{\Delta t}=B(1)\;\times \;F-K_{a}\times \;X_{1}$$
(2)$$\frac{\Delta X_{2}}{\Delta t}=K_{a}\times \;X_{1}-K_{e}\;\times \;X_{2}$$

Equations (1) and (2) describe the rate of change of the ACV amount in the compartments. X_1_ is the ACV amount in the gut (mg) and X_2_ is the ACV amount in the central compartment (mg). B(1) is the input of VCV. F is the bioavailability of VCV. K_a_ is the absorption rate constant (h^−1^). K_e_ is the elimination rate constant (h^−1^).

The following equation was used for the PK/PD model [20]:(3)$$\frac{\Delta X_{3}}{\Delta t}=\mathrm{initial}\;\mathrm{condition}-K_{\mathrm{kmax}}\times \frac{{\left(\frac{X_{2}}{V}\right)}^{\mathrm{Hk}}}{{\left(\frac{X_{2}}{V}\right)}^{\mathrm{Hk}}+{\mathrm{EC}}_{50}{}^{\mathrm{Hk}}}\times X_{3}$$

Equation (3) describes the change in VZV viral load in plasma (log_10_ viral copies/mL). The initial condition (IC) is the VZV viral load (viral copies/mL) at time = 0. K_kmax_ is the maximal rate of drug-induced viral kill (log_10_ viral copies/mL/h). X_2_/V is the ACV concentration in plasma (mg/L). The EC_50_ is the concentration of acyclovir that induces 50% of the maximal rate of kill (mg/L). Hk is the slope function. The parameter distribution used for prediction in mean and median was tested.

## 3. Results

### 3.1. Patient Population

The PK/PD model was fitted to immunocompromised patients with both ACV levels and VZV plasma viral loads (*N* = 4 with 23 measure points). The patient characteristics are shown in Table 1. These 4 patients were part of a heterogeneous population who were included for PK model building (*N* = 43 with 96 ACV concentration data points). Patient characteristics of this population (*N* = 43) are described in Appendix A.

### 3.2. PK Modelling

The final PK model was a one-compartment model with linear elimination without covariates, since a two-compartment model, as well as including covariates, did not improve the model. The PK parameters are described in Table 2. The final model adequately described the patient data. The individual posterior prediction goodness-of-fit plot resulted in an r^2^ = 0.95 (Appendix A). The individual PK parameters of the final PK model were used as covariates in the PK/PD model.

### 3.3. PK/PD Modelling

PD parameters of the PK/PD model are shown in Table 3. The individual posterior prediction goodness-of-fit plot resulted in an r^2^ = 0.83 (Figure 1a). The individual VZV viral load vs. time plots (Figure 1b) showed that the predicted viral load curve is comparable with the observed viral loads. 

## 4. Discussion

Currently, there is no clear understanding of the pharmacodynamics of acyclovir. Therefore, it is important to develop a PK/PD model to evaluate the dose–concentration–response relationship for optimal treatment. To our knowledge, this is the first study that describes a PK/PD model of ACV in the treatment of VZV. First, a PK model was created to describe and predict ACV levels. The effect of ACV concentration on viral load was investigated by developing a PK/PD model. Some studies have already given a small insight into the pharmacodynamics of acyclovir. Safrin et al. concluded that there is a strong association between EC_50_ and clinical response after treatment of HSV-infected immunocompromised patients [5]. Moreover, IC_50_ values are already incorporated in dosing guidelines: a higher ACV or VCV dose is recommended to treat VZV than to treat HSV [2], since the IC_50_ of VZV is higher [3]. Furthermore, a significant dose–response relationship has previously been found for 1 × 250 mg VCV, 1 × 500 mg VCV and 1 × 1000 mg VCV for suppression of recurrent genital herpes, suggesting a concentration–response relationship [16]. However, the efficacy of ACV was associated with time-dependent killing in a study where ACV 400 mg twice daily was shown to be significantly more effective (lower rate of recurrence of genital herpes) than VCV 500 mg once daily, despite ACV maximal concentration (C_max_) and area under the curve (AUC) being higher for VCV 500 mg once daily [16].

Our results are limited by the small sample size: only four patients were included in the PK/PD model. However, we used all available data in The Netherlands. No other hospital in the Netherlands we contacted measured ACV levels. The PK/PD model was derived from a similar modelling study on ganciclovir and cytomegalovirus conducted in our department [20]. Another drawback of this study is that the converted viral loads in log_10_ copies/mL are an estimate of the VZV viral load, since the calibration line of EBV was used. However, we used the relative differences between viral loads and, therefore, the reliability of our PK/PD model is unlikely to be affected.

Despite these limitations, our PK/PD model provides a good first step in clarifying the pharmacodynamics of ACV, inviting other researchers to study the effect of ACV levels on VZV viral load. In this way, evidence-based PD targets can be determined to improve exposure and potentially accelerate viral clearance. We would like to call on hospitals internationally to share available data. 

## Figures and Tables

**Figure 1 pharmaceutics-14-02311-f001:**
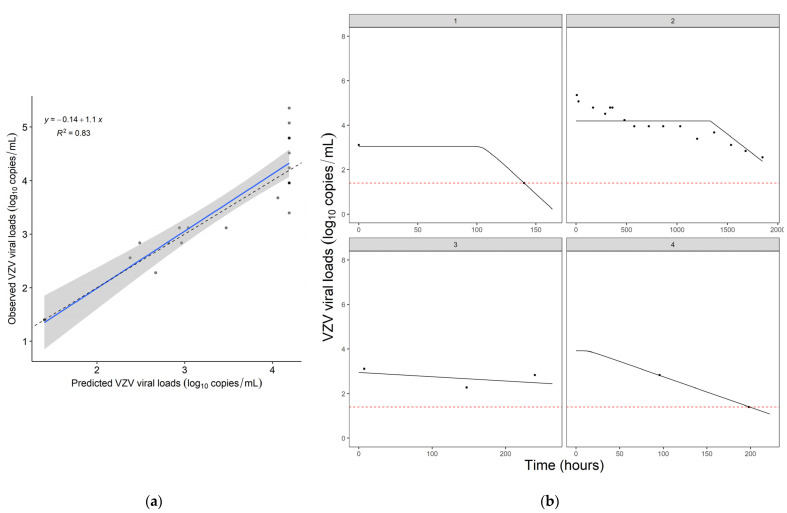
Results of PK/PD model of the effect of acyclovir on VZV viral loads in plasma (*N* = 4) (**a**) Individual posterior predictions goodness-of-fit plot of mean of final PK/PD model. The solid line represents the linear regression of the observed and predicted viral loads, the dashed line represents a reference line for y = x and the black dots display the observed VZV viral loads (**b**) Individual VZV viral load (log_10_ copies/mL) vs. time (hours) plots. The black dots represent the observed viral loads, the black line displays the individual predicted plot and the dotted line is the limit of detection. PK: Pharmacokinetic; PD: pharmacodynamic; VZV: varicella zoster virus.

**Table 1 pharmaceutics-14-02311-t001:** Patient characteristics for pharmacokinetic/pharmacodynamic modelling (*N* = 4). Mean AUC_24 h_ is calculated between first and last viral load measurement.

Patient Number	Age (Years)	eGFR (mL/min/1.73 m^2^)	First Measured VZV Viral Load (Log_10_ Copies/mL)	Last Measured VZV Viral Load (Log_10_ Copies/mL)	Mean AUC_24h_ (h × mg/L)
1	2	143	3.1	Undetectable	29
2	60	131	5.4	2.6	66
3	36	26	3.1	2.8	215
4	69	73	2.8	Undetectable	120

VZV: varicella zoster virus; AUC_24h_: area under the curve for 24 h; eGFR: estimated glomerular filtration rate.

**Table 2 pharmaceutics-14-02311-t002:** PK parameters of final PK model (*N* = 43).

PK Parameter	Mean	Median	SD	Shrinkage
Ke (h^−1^)	0.274	0.235	0.191	43.0%
V (L)	44.298	52.173	16.351	47.6%
Ka (h^−1^)	0.601	0.557	0.385	59.1%
FA	0.443	0.438	0.226	52.3%

PK: pharmacokinetic; K_e_: elimination rate constant; V: volume of distribution; K_a_: absorption rate constant; FA: bioavailability of valacyclovir; SD: standard deviation.

**Table 3 pharmaceutics-14-02311-t003:** PD parameters of final PK/PD model (*N* = 4).

PD Parameter	Mean	Median	SD
K_kmax_ (log_10_ copies/mL/h)	0.040	0.020	0.044
Hk	6.29	3.27	5.29
IC (copies/mL)	6442	4661	6029
EC_50_ (mg/L)	10.50	8.37	10.91

PD: pharmacodynamic; K_kmax_: the maximal rate of drug-induced viral kill; Hk: the slope function; IC: initial condition; EC_50_: acyclovir concentration inducing half maximal rate of kill; SD: standard deviation.

## Data Availability

Not applicable.

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
