# Peer review of "Shedding a Light on Acyclovir Pharmacodynamics: A Retrospective Analysis on Pharmacokinetic/Pharmacodynamic Modelling of Acyclovir for the Treatment of Varicella Zoster Virus Infection in Immunocompromised Patients: A Pilot Study"

_pharmaceutics, 2022, doi:10.3390/pharmaceutics14112311_

Round 1
Reviewer 1 Report
As the authors point out, there is currently no clear understanding of the pharmacodynamics of acyclovir, a drug with very unfavorable bioavailability potential after oral administration. The development of a model evaluating the dose/concentration relationship allows for effective optimization of the dose used in a patient. The description of the PK/PD model for ACV in the treatment of VZV is therefore very important. It is worth noting that studies were undertaken with the participation of a group of immunocompromised patients, and therefore a group particularly susceptible to infections and their consequences. The authors describe the advantages and disadvantages of their own research, citing a critical discussion and urging hospitals to cooperate in order to optimize patient therapy, emphasizing the essence and need to create PK/PD models in clinical practice.
I do not see any flaws in the research methodology used by the authors and their interpretation. However, at this stage of research (small population), is it possible to formulate conclusions about the real usefulness of the developed model? Nevertheless, I consider the work very important from the point of view of therapy, especially of immunocompromised patients.
Reviewer 2 Report
Dear authors,
the report is well conducted. But if in the title there is reported " Immunocompromised patients", in ABSTRACT (and in particular in "results" part), you have to highlight the 4 patients first and then say that they are part of a larger population, because the model was developed on them. Instead, as it is now, it seems to have developed on 43 patients. Risks to mislead the reader.In "discussion" you reported correctly.
I would add "retrospective analysis" in title..it is more correct and immediately gives an idea of the type of study that the reader is going to read.
Reviewer 3 Report
The manuscript presented deals with an important subject and may be accepted for publication after some remarks have been corrected.
Thus, in the introduction the IC50 data for acyclovir and valacyclovir are presented and a reference to the review article 3 is given. In addition, it is more correct to cite the original studies rather than the review article.
On line 10 the authors write - The PK parameters are described in Table S2
in the reviewer's opinion this information is important and it makes sense to move the table to the main text and not to the supplementary material.
Figure 1a shows a table, it should be shown separately as a table and not as part of the figure. The quality of the Figures should be improved.
The main criticism relates to figure 1c. Judging by the Figure, the applicability of the model is doubtful as there are too few experimental data points for patients 1, 3 and 4. Two or three data points can be fitted by any function. Moreover, the pattern of the data points for participant 2 looks more like to be fitted by a linear function rather than the offered graph.
